# Isolation and Characterization of *Euglena gracilis*-Associated Bacteria, *Enterobacter* sp. CA3 and *Emticicia* sp. CN5, Capable of Promoting the Growth and Paramylon Production of *E. gracilis* under Mixotrophic Cultivation

**DOI:** 10.3390/microorganisms9071496

**Published:** 2021-07-13

**Authors:** Kazuhiro Mori, Daisuke Inoue, Sunah Kim, Jaecheul Yu, Taeho Lee, Michihiko Ike, Tadashi Toyama

**Affiliations:** 1Integrated Graduate School of Medicine, Engineering, and Agricultural Sciences, University of Yamanashi, 4-3-11 Takeda, Kofu 400-8511, Yamanashi, Japan; g18dtka4@yamanashi.ac.jp; 2Graduate Faculty of Interdisciplinary Research, University of Yamanashi, 4-3-11 Takeda, Kofu 400-8511, Yamanashi, Japan; mori@yamanashi.ac.jp; 3Division of Sustainable Energy and Environmental Engineering, Graduate School of Engineering, Osaka University, 2-1 Yamadaoka, Suita 565-0871, Osaka, Japan; d.inoue@see.eng.osaka-u.ac.jp (D.I.); ike@see.eng.osaka-u.ac.jp (M.I.); 4Department of Civil and Environmental Engineering, Pusan National University, Busan 46241, Korea; sunah.kim2401@pusan.ac.kr (S.K.); yjcall0715@pusan.ac.kr (J.Y.); leeth55@pusan.ac.kr (T.L.)

**Keywords:** *Euglena gracilis*, paramylon, associated bacteria, growth promotion, paramylon production promotion, co-culture, *Emticicia* sp., *Enterobacter* sp.

## Abstract

*Euglena gracilis* produces paramylon, which is a feedstock for high-value functional foods and nutritional supplements. The enhancement of paramylon productivity is a critical challenge. Microalgae growth-promoting bacteria (MGPB) can improve microalgal productivity; however, the MGPB for *E. gracilis* remain unclear. This study isolated bacteria capable of enhancing *E. gracilis* growth and paramylon production under mixotrophic conditions. *Enterobacter* sp. CA3 and *Emticicia* sp. CN5 were isolated from *E. gracilis* grown with sewage-effluent bacteria under mixotrophic conditions at pH 4.5 or 7.5, respectively. In a 7-day *E. gracilis* mixotrophic culture with glucose, CA3 increased *E. gracilis* biomass and paramylon production 1.8-fold and 3.5-fold, respectively (at pH 4.5), or 1.9-fold and 3.5-fold, respectively (at pH 7.5). CN5 increased *E. gracilis* biomass and paramylon production 2.0-fold and 4.1-fold, respectively (at pH 7.5). However, the strains did not show such effects on *E. gracilis* under autotrophic conditions without glucose. The results suggest that CA3 and CN5 promoted both *E. gracilis* growth and paramylon production under mixotrophic conditions with glucose at pH 4.5 and 7.5 (CA3) or pH 7.5 (CN5). This study also provides an isolation method for *E. gracilis* MGPB that enables the construction of an effective *E. gracilis*–MGPB-association system for increasing the paramylon yield of *E. gracilis.*

## 1. Introduction

Microalgal biomass is a sustainable and renewable feedstock for value-added products, such as feeds and foods, food supplements, pharmaceuticals, personal care products, and biofuels [1]. Among the microalgal species, *Euglena gracilis*, a freshwater unicellular flagellated microalga, is among the most promising industrially applicable microalga for producing commercial sources of high-value-added products [2]. *E. gracilis* can grow photoautotrophically, heterotrophically, and mixotrophically [3,4,5,6] and accumulate various metabolites, such as amino acids [6], vitamins [7], polyunsaturated fatty acids [8], and paramylon (β-1,3-glucan) [9].

Paramylon has unique and medical functions, such as immunomodulation [10,11], anti-allergy [12], cholesterol-lowering [13], anti-tumor [14], anti-HIV [12], and anti-infection activities [15]. Paramylon is thus recognized as a feedstock for high-value functional foods and nutritional supplements. The paramylon industry has received significant attention [2], and its market is expected to increase in the future. *E. gracilis* can grow and accumulate paramylon under both low and neutral pH conditions [9]. The cultivation of *E. gracilis* under acidic conditions gives it an important advantage over other microalgae, as it prevents microbial and bacterial contaminations and enables large-scale cultivation for the paramylon industry. Enhancing the growth and paramylon productivity of *E. gracilis* is the most important challenge to realizing highly efficient *E. gracilis*-based industrial paramylon production.

In natural aquatic environments and microalgal cultures, microalgae often co-exist with symbiotic bacteria. Certain symbiotic bacteria [termed microalgae growth-promoting bacteria (MGPB)] can promote microalgal growth in various ways, such as by providing nutrients [16], vitamins [17], phytohormones [18,19], chelators [20], and volatile organic compounds [21]. MGPB associated with several microalgae species have been isolated from microalgal host cultures. For example, a *Rhizobium* sp. strain was isolated from *Chlorella vulgaris* [22], a *Rhizobium* sp. strain was isolated from *Botryococcus braunii* [23], the *Candidatus* Phycosocius bacilliformis strain BOTRYCO-2 was isolated from *B. braunii* [24], *Pelagibaca bermudensis* KCTC 13073BP and *Stappia* sp. KCTC 13072BP were isolated from *Tetraselmis striata* [25], a *Pseudomonas* sp. was isolated from *Scenedesmus* sp. LX1 [19,26], and a *Marinobacter* sp. was isolated from *Chlamydomonas reinhardtii* [22]. Only one MGPB (*Emticicia* sp. strain EG3) has been isolated from the cell surface of *E. gracilis*, and it promoted *E. gracilis* growth under photoautotrophic conditions [27], although the promotion of *E. gracilis* paramylon production by EG3 was unclear. Thus, little information is available regarding *E. gracilis*-associated MGPB and their ability to promote *E. gracilis* paramylon production.

By contrast, previous studies demonstrated that indole-3-acetic acid (IAA)-producing *Vibrio natriegens* [28] and extracellular polymeric substance (EPS)-producing *Pseudoalteromonas* sp. MEBiC 03485 [29] and *Pseudoalteromonas* sp. MEBiC 03607 [30] improved the growth and paramylon productivity of *E. gracilis* under heterotrophic conditions. Additionally, strains MEBiC 03485 and MEBiC 03607 increased the expression of genes related to β-1,3-glucan synthase [29,30]. However, these three bacteria were not isolated from *E. gracilis*-associated environments, but rather they were obtained from microbial culture collection centers. Thus, these bacteria do not have any inherent advantage in associating with *E. gracilis* cells and building a symbiotic relationship. Screening and isolating MGPB symbiotically associated with *E. gracilis* enable the building of a more effective, sustainable, and practical *E. gracilis*–bacteria association for increasing paramylon production and contributing to the *E. gracilis*-based paramylon industry. However, to date, no MGPB that enhance both *E. gracilis* growth and paramylon production have been isolated from the host *E. gracilis*.

Therefore, the objectives of this study were to screen, isolate, and characterize MGPB capable of enhancing both *E. gracilis* growth and paramylon production under mixotrophic conditions. Previous studies reported that *E. gracilis* showed greater paramylon production and growth during mixotrophic cultivation as compared with photoautotrophic conditions [3,5,6]; therefore, we conducted experiments under mixotrophic conditions with a 12-h light and 12-h dark cycle. Because we recently observed that indigenous bacterial communities in sewage effluent promotes *E. gracilis* growth [31], we attempted to isolate MGPB from *E. gracilis* grown in sewage effluent. First, *E. gracilis* was cultured with a sewage-effluent bacterial community under mixotrophic conditions at an acidic pH (4.5) or a neutral pH (7.5), after which *E. gracilis*-associated bacteria were isolated as MGPB candidates. Second, MGPB showing growth- and paramylon-production-promoting effects on *E. gracilis* were screened and characterized. Finally, MGPB showing growth- and paramylon-production-promoting effects on *E. gracilis* were examined in *E. gracilis*–MGPB co-culture experiments.

## 2. Materials and Methods

### 2.1. Microalgae and Culture Conditions

Axenic (bacteria-free) *E. gracilis* (NIES-48) was obtained from the Microbial Culture Collection at the National Institute for Environmental Studies (NIES Collection; Tsukuba, Japan) and cultured in C medium supplemented with 400 mg L^−1^ yeast extract and 600 mg L^−1^ polypeptone (referred to here as CYP medium), as recommended by NIES Collection. Each liter of C medium contained 150 mg Ca(NO_3_)_2_·4H_2_O, 100 mg KNO_3_, 50 mg β-Na_2_glycerophosphate·5H_2_O, 40 mg MgSO_4_·7H_2_O, 500 mg Tris (hydroxymethyl) aminomethane, 0.1 μg vitamin B12, 0.1 μg biotin, 10 μg thiamine HCl, and 3 mL PIV metals (1000 mg L^−1^ Na_2_EDTA·H_2_O, 196 mg L^−1^ FeCl_3_·6H_2_O, 36 mg L^−1^ MnCl_2_·4H_2_O, 10.4 mg L^−1^ ZnCl_2_, 4 mg L^−1^ CoCl_2_·6H_2_O, and 2.5 mg L^−1^ Na_2_MoO_4_·H_2_O). The pH of the CYP medium was adjusted to 7.5. *E. gracilis* cells were sub-cultured by transferring them to fresh CYP medium every 1 or 2 weeks in a growth chamber maintained at 28 ± 1 °C with fluorescent lamps at a photosynthetic photon-flux density of 80 μmol m^−2^ s^−1^ and a 12-h photoperiod. Because *E. gracilis* use ammonium as a nitrogen source, for each experiment, axenic *E. gracilis* cells were pre-cultured in C-NH_4_ medium comprising C medium with 100 mg L^−1^ (NH_4_)_2_SO_4_ at a pH 4.5 or 7.5, depending on the experiment. Pre-cultured axenic *E. gracilis* cells were incubated in the growth chamber for 1 week.

### 2.2. Sewage Effluent Sample and Collection of Indigenous Bacterial Communities from the Effluent

Sewage effluent was collected from the final sedimentation tank of a conventionally activated sludge process of a sewage-treatment plant in Kofu City, Yamanashi, Japan. The water quality of the effluent was 11.8 mg L^−1^ of total organic carbon, 6.5 mg L^−1^ NH_4_-N, 0.08 mg L^−1^ NO_2_-N, 4.2 mg L^−1^ NO_3_-N, and 3.2 mg L^−1^ PO_4_-P. The effluent sample was first passed through a glass microfiber filter (pore size, 1.6 μm; GF/A grade; GE Healthcare UK, Ltd., Little Chalfont, UK) to remove suspended solids and organisms larger than bacteria (including microalgae) from the effluent sample. Therefore, the effluent filtrate included an indigenous bacterial community. To collect the indigenous bacteria, the effluent sample (300 mL) was filtered through a sterilized membrane filter (pore size, 0.2 µm; polytetrafluoroethylene; Merck Millipore Ltd., Cork, Ireland), which were then placed into 30 mL of C medium in a 50-mL tube, vortexed at maximum speed for 1 min, shaken at 150 rotations/min (rpm) for 60 min, and vortexed at maximum speed for 1 min to detach the bacterial cells from the filter and suspend them in C medium. The number of culturable bacteria in the bacterial suspension was quantified using R2A agar plates (0.5 g L^−1^ peptone, 0.5 g L^−1^ yeast extract, 0.5 g L^−1^ casamino acid, 0.5 g L^−1^ glucose, 0.5 g L^−1^ soluble starch, 0.3 g L^−1^ K_2_HPO_4_, 0.05 g L^−1^ MgSO_4_·7H_2_O, and 0.3 g L^−1^ sodium pyruvate (pH 7.0), and agar 15 g L^−1^). The culturable bacterial density was 5.7 × 10^5^ CFU mL^−1^.

### 2.3. Culturing E. gracilis with a Bacterial Community Derived from Sewage Effluent and Isolating Bacteria Associated with E. gracilis

Corn steep liquor (CSL; Oji Cornstarch Co., Ltd., Tokyo, Japan) was used as an organic carbon source for the mixotrophic cultures. Ten milliliters of *E. gracilis* (pre-cultured in C-NH_4_ medium with 0.5 g L^−1^ CSL at pH 4.5 or 7.5) was added to 100 mL of C-NH_4_ medium with 0.5 g L^−1^ CSL at pH 4.5 or 7.5 in a 200-mL glass flask, into which 10 mL of the sewage-effluent bacterial suspension was inoculated. The *E. gracilis* sewage-effluent bacterial cultures in C-NH_4_ with CSL at pH 4.5 or 7.5 were incubated in a growth chamber with shaking at 150 rpm for 10 d. Additionally, *E. gracilis* cultures in C-NH_4_ medium with 0.5 g L^−1^ CSL at pH 4.5 or 7.5 but without sewage-effluent bacteria were prepared and incubated as bacteria-free control cultures. The experiments were conducted in triplicate. After 10 d of culture, 25 mL of the *E. gracilis*-effluent bacterial culture was transferred into a 50-mL tube and vortexed at maximum speed for 3 min to disperse the *E. gracilis* and bacterial cells. The sample was then filtered through a GF/A glass microfiber filter to remove *E. gracilis* cells. The filtrate containing bacteria was serially diluted and spread on R2A agar plates with a pH of 4.5 or 7.5, which were incubated at 28 °C for 2 weeks. In this study, bacteria obtained from cultures at pH 4.5 were defined as acidophilic bacteria, and those obtained from cultures at pH 7.5 were defined as neutrophilic bacteria. Eight acidophilic bacteria and 15 neutrophilic bacteria were isolated, and a pure culture of each strain was maintained on R2A agar at pH 4.5 or 7.5. Because CSL reportedly increases the biomass and paramylon productivity of *E. gracilis* [9], CSL was initially used as an organic carbon source for the mixotrophic cultures of *E. gracilis*; however, CSL is a mixture of organic compounds. For detailed examination, we used pure glucose as the organic carbon source for *E. gracilis* mixotrophic culture in subsequent experiments.

### 2.4. Screening of MGPB Capable of Enhancing Both E. gracilis Growth and Paramylon Production

Each isolated bacterial strain was cultured in R2A liquid medium (pH 4.5 or 7.5) at 28 °C and with shaking (150 rpm) until the late logarithmic growth phase. The cells were harvested by centrifugation (10,000× *g*, 24 °C, and 5 min) and washed twice with C-NH_4_ medium (pH 4.5 or 7.5). Each acidophilic or neutrophilic bacterial cell culture was inoculated into *E. gracilis* C-NH_4_ medium at pH 4.5 or 7.5 with 5 g L^−1^ glucose at an optical density at 600 nm (OD_600_) of 0.05. The co-cultures of *E. gracilis* with each isolated bacterial strain were incubated in a growth chamber with shaking at 150 rpm for 7 d. The initial biomass of *E. gracilis* was approximately 40 mg dry weight L^−1^. On day 7, concentrations of *E. gracilis* chlorophyll *a* + *b*, biomass, and paramylon were measured. Control cultures including *E. gracilis* cells alone (without bacterial inoculation) in C-NH_4_ medium (pH 4.5 or 7.5) were also prepared and analyzed similarly for comparison. The *E. gracilis* growth- and paramylon-production-promoting abilities of the isolated bacteria were assessed by comparing the biomass and paramylon concentrations at the end of 7-d culturing with bacterial inoculation relative to control cultures.

### 2.5. Identification and Characterization of the CA3 and CN5 Strains

Among the isolated bacterial strains, strains CA3 (acidophilic bacterium) and CN5 (neutrophilic bacterium) showed the highest growth- and paramylon-production-promoting abilities at pH 4.5 and 7.5, respectively. Strains CA3 and CN5 were characterized and identified using physiological and phylogenetic analyses. Physiological characterization was performed using an API 20NE Kit (BioMérieux Japan, Tokyo, Japan) according to the manufacturer’s instructions. Comparative 16S rRNA gene-sequence analysis was performed, as follows: almost full-length 16S rRNA genes were amplified by PCR using the primers 8F (5′-AGAGTTTGATCCTGGCTCAG-3′) and 1510R (5′-GGTTACCTTGTTACGACTT-3′). Genus-level identifications were carried out based on 16S rRNA gene-sequence similarities with those of type-strain sequences in NCBI GenBank using BLAST. The 16S rRNA sequence data [1431 base pairs (bp)] of CA3 and CN5 were submitted to the DDBJ/EMBL/GenBank databases under accession numbers LC604062 and LC604063, respectively.

Bacterial IAA production was evaluated as described previously [32], with some modifications. The bacterial colony was inoculated in 100 mL of C-NH_4_ medium with 5 g L^−1^ glucose (pH 4.5 or 7.5) with or without 0.05% (*w*/*v*) l-tryptophan and incubated at 28 °C and 150 rpm for 1 d. The culture was collected and centrifuged (10,000× *g*, 4 °C, and 10 min), and 500 μL supernatant was mixed with 750 μL Salkowski reagent (98 mL 35% HClO_4_ plus 2 mL of 0.5 M FeCl_3_) and incubated at 24 °C for 30 min. The development of a pink color indicated IAA production, and the absorbance at 535 nm (A_535_) was measured. The IAA concentration was calculated using pure IAA as a standard (Kanto Chemical Co., Inc., Tokyo, Japan).

Bacterial EPS production was evaluated as described previously [29], with specific modifications. Bacterial colonies were inoculated in 200 mL C-NH_4_ medium with 5 g L^−1^ glucose (pH 4.5 or 7.5) and incubated at 28 °C and 150 rpm for 1 d. Each culture was collected and centrifuged (10,000× *g*, 4 °C, and 20 min), and the supernatant was gently mixed with three volumes of ice-cold 100% ethanol and incubated overnight at 4 °C. The precipitated EPS was collected by centrifugation (10,000× *g*, 4 °C, and 20 min) and dried. This EPS was defined as free EPS. Bacterial cells were collected from the above 1-d culture by centrifugation (10,000× *g*, 24 °C, and 20 min) and mixed with 0.9% NaCl solution, with this mixture homogenized and shaken using a vortex for 3 min and centrifuged (10,000× *g*, 4 °C, and 20 min). The supernatant was gently mixed with three volumes of ice-cold 100% ethanol and incubated at 4 °C overnight. The precipitated EPS was collected by centrifugation (10,000× *g*, 4 °C, and 20 min), dried, and defined as cell-bound EPS. Free and cell-bound EPS were dissolved in hot distilled water, and the total sugar content of the EPS was determined in each sample by the phenol–sulfuric acid method [33]. The total protein content of the EPS was estimated in each sample using a BCA protein assay kit (Takara Bio, Shiga, Japan). Total sugar and protein contents in the EPS were calculated as mg L^−1^ of culture medium.

### 2.6. Growth of CA3 and CN5 Utilizing Glucose

Strain CA3 was pre-cultured at 28 °C in liquid R2A medium (pH 4.5) with shaking (150 rpm) until it reached the late logarithmic growth phase. Similarly, strain CN5 was pre-cultured at 28 °C in R2A medium (pH 7.5) with shaking (150 rpm) un2w1til it reached the late logarithmic growth phase. The cells of each strain were harvested by centrifugation (10,000× *g*, 24 °C, and 5 min) and washed twice with C-NH_4_ medium (pH 4.5 or 7.5). The cells were inoculated into 200 mL C-NH_4_ medium with 5 g L^−1^ glucose (pH 4.5 or 7.5) in a 500-mL flask until reaching an OD_600_ of 0.05, after which they were incubated for 5 d at 28 °C and 150 rpm in the dark. The bacterial cell densities (OD_600_) and glucose concentrations were monitored during the incubation period. The growth experiments were conducted in triplicate.

### 2.7. Co-Culturing E. gracilis with CA3 or CN5 under Acidic and Neutral pH with or without Glucose

Ten milliliters of *E. gracilis* pre-cultured in C-NH_4_ medium with 5 g L^−1^ glucose (pH 4.5 or 7.5) was inoculated into 200 mL C-NH_4_ (pH 4.5 or 7.5) with or without 5 g L^−1^ glucose. The initial biomass of *E. gracilis* was approximately 40 mg dry weight L^−1^. CA3 or CN5 cells were pre-cultured in liquid R2A medium at 28 °C with shaking (150 rpm) at pH 4.5 or 7.5, respectively, until they reached late logarithmic growth phase. CA3 or CN5 cells were harvested by centrifugation (10,000× *g*, 24 °C, and 5 min) and washed twice with C-NH_4_ medium (pH 4.5 or 7.5). CN3 or CN5 cells were inoculated at an OD_600_ of 0.05 into *E. gracilis* C-NH_4_ medium at pH 4.5 or 7.5 with or without 5 g L^−1^ glucose. Control cultures including *E. gracilis* cells alone (without bacterial inoculation), CA3 or CN5 cells alone (without *E. gracilis*), and C-NH_4_ medium with 5 g L^−1^ glucose alone (without *E. gracilis* and bacterial cultures) were also prepared. All culture flasks were incubated in a growth chamber with shaking at 150 rpm for 7 d. During the experiments, chlorophyll *a* + *b*, *E. gracilis* biomass, paramylon, and glucose concentrations were monitored on the initial day and days 3 and 7. The experiments were conducted in triplicate. The growth- and paramylon-production-promoting effects of CA3 and CN5 were evaluated for comparison with *E. gracilis* axenic control culture in 7-d cultures.

### 2.8. Scanning Electron Microscopy (SEM) of E. gracilis Cell Surfaces

For SEM experiments, *E. gracilis* cells were collected from the co-culture experiment at 7 d and centrifuged at 3000× *g* for 5 min at 24 °C. Additionally, the collected *E. gracilis* cells were washed using the same biomass (dry weight)-measurement method: vortex mixing for 60 s, centrifugation (3000× *g*, 24 °C, and 5 min), and washing with distilled water. The *E. gracilis* cells (with or without the above washing process) were then fixed with 4% osmium tetroxide solution at 4 °C for 3 h, dehydrated at room temperature (~24 °C) in solutions containing progressivelyincreasing ethanol concentrations (30–100%; 15 min/incubation), and finally dried at the carbon dioxide critical point. The dried samples were coated using an osmium plasma coater (OPC80T; Filgen, Nagoya, Japan) and then examined by SEM using a JEOL SEM instrument (JSM 6320F; JEOL Ltd., Tokyo, Japan).

### 2.9. Analytical Procedures

The chlorophyll concentrations in the *E. gracilis* cultures were measured spectrophotometrically after extraction with 100% methanol for 30 min [34]. The A_665_ and A_650_ of each extract were measured using a UVmini-1240 spectrophotometer (Shimadzu Co. Ltd., Kyoto, Japan). Total chlorophyll (Chl; Chl *a* + *b*) concentration (μg mL^−1^) was calculated, as follows:Chl *a* + *b* (μg mL^−1^) = (4 × A_665_) + (25.5 × A_650_)(1)

The biomass (dry weight) of *E. gracilis* samples was measured, as follows: 25 mL of each culture was collected into a 50-mL centrifuge tube and vortexed for 60 s to uniformly suspend bacterial and microalgal cells. Each mixture was centrifuged (3000× *g*, 24 °C, and 5 min), the pellet was washed with 25 mL distilled water, and the centrifugation and wash steps were repeated one time. Subsequently, each *E. gracilis* pellet was suspended in 25 mL distilled water, collected on a pre-weighed GF/A glass fiber filter, dried at 90 °C for 3 h, and then weighed. We confirmed by SEM observations that the collected *E. gracilis* cells contained no CA3 or CN5 cells.

Paramylon was extracted from *E. gracilis* cells using the sodium dodecyl sulfate–ethylenediaminetetraacetic acid (SDS–EDTA) method. Each *E. gracilis* culture (25 mL) was collected into a 50-mL centrifuge tube, ultrasonicated for 1 min, vortexed for 30 s, and then centrifuged (3000× *g*, 24 °C, and 5 min) to remove the supernatant. The collected *E. gracilis* pellets were washed thrice by centrifugation (3000× *g*, 24 °C, and 5 min), resuspended in 25 mL distilled water, and incubated with 10 mL of ethanol for 30 min at room temperature (24 °C) before another round of centrifugation (8000× *g*, 24 °C, and 5 min). The collected *E. gracilis* cells were mixed with 10 mL SDS–EDTA reagent (1% SDS: 5% Na_2_·EDTA), incubated in a water bath at 90 °C for 30 min, and centrifuged (8000× *g*, 24 °C, and 5 min), after which the supernatant was removed. Each collected *E. gracilis* pellet was mixed with 1 mL SDS–EDTA reagent and 9 mL distilled water, vortexed, and centrifuged (8000× *g*, 24 °C, and 5 min), after which each supernatant was removed from the tube. The collected *E. gracilis* pellets were then mixed with 20 mL distilled water, vortexed, and centrifuged (8000× *g*, 24 °C, and 5 min), after which each supernatant was removed from the tube. Each collected *E. gracilis* pellet was then mixed with 5 mL 1 mol L^−1^ NaOH, vortexed, and incubated at room temperature (24 °C) for 12 h. A 0.5-mL aliquot from each tube was transferred to a test tube with 0.5 mL of 5% phenol solution and 2.5 mL sulfuric acid. The test tubes were gently mixed and incubated at room temperature (24 °C) for 30 min, after which the A_480_ of each solution was measured using a spectrophotometer. A standard calibration curve was prepared using commercially available 100% paramylon (β-1,3-glucan from *E. gracilis*; Sigma-Aldrich, St. Louis, MO, USA).

The glucose concentration in each culture was measured using a Shimadzu high-performance liquid chromatography system (Shimadzu Co. Ltd.) with a refractive index detector and a Shodex SUGAR SH1011 column (300 mm × 8.0 mm; Showa Denko K. K., Tokyo, Japan). The mobile phase was 1 mmol L^−1^ sulfuric acid solution, and the column was maintained at 50 °C.

### 2.10. Statistical Analysis

Each value presented represents the results of three replicates (*n* = 3) per experiment. All results are expressed as the mean ± standard deviation (SD). Statistical significance (*p* < 0.05) was analyzed using the paired-samples *t*-test with SPSS Statistics (v.22.0; IBM Corp., Armonk, NY, USA).

## 3. Results

### 3.1. Isolation and Identification of Bacteria Promoting the Growth and Paramylon Production of E. gracilis

*E. gracilis* cells were cultured in C-NH_4_ with 0.5 g L^−1^ CSL under mixotrophic conditions at pH 4.5 or 7.5 with or without sewage-effluent bacteria. *E. gracilis* clearly showed faster growth at both pH values when co-cultured with sewage-effluent bacteria than without (Appendix A). Therefore, this suggested that MGPB for *E. gracilis* must have been present among the sewage-effluent bacteria and supported *E. gracilis* growth under mixotrophic conditions at pH 4.5 or 7.5.

After *E. gracilis* cells were co-cultured with sewage-effluent bacteria at pH 4.5 or 7.5, eight acidophilic bacterial strains and 15 neutrophilic bacterial strains were isolated, respectively (Appendix A). All isolates showed at least 95% sequence identity with known type strains, as determined by BLAST searches for the partial 16S rDNA sequences of the isolated strains (Appendix A). Among the isolated acidophilic bacteria strains, three (CA1, CA2, and CA7) were *Microbacterium* sp., three (CA6, CA8, and CA9) were *Achromobacter* sp., one (CA4) was a *Kaistia* sp., and one (CA3) was an *Enterobacter* sp. Among the isolated neutrophilic bacterial strains, three (CN1, CN10, and CN11) were *Pedobacter* sp., one (CN2) was a *Phenylobacterium* sp., one (CN3) was a *Sphingomonas* sp., one (CN4) was an *Achromobacter* sp., one (CN5) was an *Emticicia* sp., one (CN6) was an *Elizabethkingia* sp., one (CN7) was an *Enterobacter* sp., one (CN8) was a *Herminiimonas* sp., three (CN9, CH13, and CN15) were *Sediminibacterium* sp., one (CN12) was an *Erythrobacter* sp., and one (CN14) was a *Polaromonas* sp. Among these strains, CA3 (an acidophilic bacteria) and CN5 (a neutrophilic bacteria) showed the highest growth- and paramylon-production-promoting effects on *E. gracilis* under mixotrophic conditions with 0.5 g L^−1^ glucose at a pH of 4.5 or 7.5, respectively (data not shown).

On R2A agar, the CA3 colonies appeared convex, circular, smooth, lustrous, and white in color and were Gram-negative and rod-shaped (0.8–1.0 × 1.2–1.8 μm) (Appendix A). CA3 tested positive for nitrate reduction, glucose fermentation, aesculin hydrolysis, and β-galactosidase activity but negative for oxidase, arginine dihydrolase, urease, and gelatinase activities. CA3 utilized d-glucose, l-arabinose, d-mannose, d-mannitol, glucosamine, d-maltose, gluconate, malate, citrate, and phenylacetate but did not utilize caprate or adipate. Almost the entire 16S rRNA gene sequence (1467 bp) of CA3 was similar to that of *Enterobacter roggenkampii* DSM 16,690^T^ [99.6% (1461/1467 bp) sequence similarity], *Enterobacter mori* LMG 25706^T^ [99.5% (1460/1467 bp)], *Enterobacter oligotrophicus* CCA6^T^ [99.5% (1460/1467 bp)], *Enterobacter wuhouensis* WCHEs120002^T^ [98.8% (1450/1467 bp)], *Enterobacter asburiae* ATCC 35953^T^ [98.8% (1450/1467 bp)], and *Enterobacter xiangfangensis* LMG27195^T^ [98.8% (1449/1467 bp)]. Based on phylogenetic analysis, CA3 was identified as an *Enterobacter* sp.

By contrast, colonies of strain CN5 on R2A agar were convex, circular, smooth, lustrous, and orange in color. Similar to the CA3 strain, the CN5 strain was also Gram-negative and rod-shaped (0.7–0.9 × 1.2–1.6 μm) (Appendix A). CA5 cells were positive for oxidase, aesculin hydrolysis, and β-galactosidase activities but negative for nitrate reduction, glucose fermentation, and arginine dihydrolase, urease, and gelatinase activities. Strain CN5 utilized d-glucose, d-mannose, glucosamine, and d-maltose but did not utilize l-arabinose, d-mannitol, gluconate, caprate, adipate, malate, citrate, or phenylacetate as the sole carbon source. Almost the entire 16S rRNA gene sequence (1431 bp) of CN5 was similar to the sequences of *Emticicia fontis* IMCC1731^T^ [97.9% (1403/1433 bp)], *Emticicia ginsengisoli* Gsoil 085^T^ (97.6% [1399/1434 bp]), *Emticicia soli* ZZ-4^T^ [97.5% (1396/1432 bp)], *Emticicia oligotrophica* GPTSA100-15^T^ [94.4% (1354/1435 bp)], *Emticicia paludis* HMF3850^T^ [94.1% (1349/1433 bp)], and *Emticicia aquatica* HMF2925^T^ [93.9% (1346/1434 bp)]. Based on phylogenetic analysis, CN5 was identified as an *Emticicia* sp.

Previous reports show that IAA- and EPS-producing bacteria promote *E. gracilis* growth and paramylon production [28,29]. Therefore, we tested the IAA- and EPS-producing activities of CA3 and CN5. CA3 and CN5 did not produce IAA in C-NH_4_ + glucose medium in the absence of l-tryptophan, which is the main precursor for the IAA-biosynthesis pathways in bacteria (Table 1). Additionally, CA3 produced EPS in C-NH_4_+glucose medium at both pH 4.5 and 7.5, whereas CN5 produced EPS in C-NH_4_ + glucose medium only at pH 7.5 (Table 1).

### 3.2. Growth of CA3 and CN5 Utilizing Glucose as a Sole Carbon Source

CA3 or CN5 were cultured in C-NH_4_ medium with 5 g L^−1^ glucose at pH 4.5 or 7.5. CA3 rapidly utilized 5 g L^−1^ glucose at both pH 4.5 and 7.5, and the bacterial cell density (OD_600_) increased in parallel with glucose uptake, reaching a stationary growth phase within 12 h (Figure 1A). CN5 rapidly utilized 5 g L^−1^ glucose at pH 7.5, and bacterial growth paralleled glucose uptake, reaching stationary phase within 12 h (Figure 1B). However, CN5 did not utilize glucose or grow at pH 4.5.

### 3.3. Co-Culturing E. gracilis with Strain CA3 or CN5 under Photoautotrophic or Mixotrophic Conditions at pH 4.5 or 7.5

*E. gracilis* cells were co-cultured with CA3 or CN5 cells in C-NH_4_ medium without glucose (photoautotrophic condition) or with 5 g L^−1^ glucose (mixotrophic condition) at pH 4.5 or 7.5 for 7 d. At pH 4.5 under photoautotrophic conditions without glucose, the chlorophyll, biomass, and paramylon concentrations of *E. gracilis* co-cultured with CA3 or CN5 and axenic control *E. gracilis* cultures increased at comparable rates, although slight differences were observed (Figure 2A–C).

*E. gracilis* biomass and paramylon concentrations were slightly but significantly higher (*p* < 0.05) after 3 d in *E. gracilis* co-cultured with CA3 than when co-cultured with CN5 or in axenic control cultures (Figure 2B,C). At pH 4.5 under mixotrophic conditions with glucose, the chlorophyll, biomass, and paramylon concentrations of *E. gracilis* increased rapidly and were significantly higher (*p* < 0.05) in *E. gracilis* co-cultured with strain CA3 than when co-cultured with CN5 or in axenic control cultures (Figure 2D–F). Moreover, glucose concentration decreased more rapidly in *E. gracilis* co-cultured with CA3 than with CN5 or in axenic control *E. gracilis* cultures (Figure 2G). After 7 d, the final biomass and paramylon concentrations of *E. gracilis* co-cultured with CA3 were 1.8- and 3.5-fold higher, respectively, as compared with those of control *E. gracilis* cultured under mixotrophic conditions with glucose (Table 2).

At pH 7.5 under photoautotrophic conditions without glucose, the chlorophyll, biomass, and paramylon concentrations of *E. gracilis* co-cultured with CA3 or CN5 and axenic control *E. gracilis* cultures increased at comparable rates, although slight differences were noted (Figure 3A–C). At pH 7.5 under mixotrophic conditions with glucose, the chlorophyll concentrations in *E. gracilis* co-cultured with CA3 or CN5 and control cultures increased at comparable rates (Figure 3D).

*E. gracilis* biomass and paramylon concentrations increased rapidly and were significantly higher (*p* < 0.05) in *E. gracilis* co-cultured with strain CA3 or CN5 than those in axenic control cultures (Figure 3E,F). Additionally, the glucose concentration in *E. gracilis* co-cultured with CA3 or CN5 decreased more rapidly than in the control *E. gracilis* culture (Figure 3G). After 7 d, the final biomass and paramylon concentrations of *E. gracilis* co-cultured with CA3 were 1.9- and 3.5-fold higher, respectively, as compared with the control *E. gracilis* culture. Moreover, after 7 d, the final biomass and paramylon concentrations of *E. gracilis* co-cultured with CN5 were 2.0- and 4.1-fold higher, respectively, as compared with the control *E. gracilis* culture under mixotrophic conditions (Table 2). In control cultures containing CA3 or CN5 alone (pH 4.5 or 7.5), *E. gracilis* biomass (collected on GF/A filters) and paramylon (extracted using the SDS–EDTA method) were not detected.

SEM observations revealed that CA3 and CN5 attached to *E. gracilis* cells in co-cultures under mixotrophic conditions at pH 4.5 or 7.5, respectively. Interestingly, CA3 and CN5 attached to the flagella of *E. gracilis* rather than the main cell surface (Figure 4A,B,D,E), although the reasons for this remain unclear. Moreover, the EPS matrix was observed at interfaces between the *E. gracilis* surface and CA3 or CN5 (Figure 4B,E). However, after the washing process, almost all CA3 and CN5 cells had detached from the *E. gracilis* surfaces (Figure 4C,F).

## 4. Discussion

In this study, *Enterobacter* sp. CA3 and *Emticicia* sp. CN5 were isolated from the surfaces of *E. gracilis* cells grown with sewage-effluent bacteria under mixotrophic conditions at pH 4.5 or 7.5, respectively. Both bacterial strains were capable of significantly promoting the growth and paramylon production of *E. gracilis* at their respective pH values during a 7-d cultivation (Figure 2 and Figure 3). Additionally, we observed that CA3 and CN5 attached to the flagella of *E. gracilis* cells and presumably formed a symbiotic association with *E. gracilis* (Figure 4A,B,D,E). *Enterobacter* sp. CA3 and *Emticicia* sp. CN5 represent the first isolated *E. gracilis*-associated bacteria capable of promoting *E. gracilis* growth and paramylon production under mixotrophic conditions.

Various *Enterobacter* spp. have been isolated from soil and water, a variety of plant species, natural animal commensals, and the human gut microbiota [35]. In the present study, *Enterobacter* sp. CA3 was successfully isolated from *E. gracilis* surfaces after growth with sewage-effluent bacteria at pH 4.5 as an MGPB. This strain is the first MGPB identified that belongs to the *Enterobacter* genus. Some *Enterobacter* spp. are acidophilic [36,37] and can grow over a wide range of pH conditions (pH 2–9) [38]. Although CA3 was isolated from *E. gracilis* grown at pH 4.5, it exhibited viability across a wide range of pH values and showed growth- and paramylon-promoting activities at both pH 4.5 and 7.5 (Figure 1, Figure 2 and Figure 3). Several *Emticicia* spp. bacteria have also been isolated from various aquatic and soil environments [39,40,41]. *Emticicia* sp. EG3 was recently isolated from *E. gracilis* grown in sewage effluent and shown to promote its growth under autotrophic and neutral pH conditions [27]. *Emticicia* sp. CN5, isolated in the present study, represents the second reported *E. gracilis*-associated MGPB, and the first *Emticicia* sp. capable of promoting paramylon production.

*Enterobacter* sp. CA3 utilized glucose for its growth (Figure 1) and significantly enhanced both the biomass and paramylon production of *E. gracilis* under mixotrophic conditions with glucose at both pH 4.5 and 7.5 but showed little or no enhancement under autotrophic conditions without glucose (Figure 2 and Figure 3). By contrast, *Emticicia* sp. CN5 utilized glucose for its growth at pH 7.5 and enhanced both *E. gracilis* biomass and paramylon production under mixotrophic conditions with glucose at pH 7.5 but did not show such enhancement under autotrophic conditions without glucose (Figure 2 and Figure 3). Interestingly, the growth- and paramylon-production-promoting effects on *E. gracilis* of the two strains were dependent on the mixotrophic condition with glucose and a viable pH condition.

*V. natriegens* produces IAA in *E. gracilis* medium with l-tryptophan and promotes the growth and paramylon production of *E. gracilis* [28]. We found that CA3 and CN5 produced IAA in C-NH_4_ + glucose medium with l-tryptophan but not when cultured in C-NH_4_+glucose medium without l-tryptophan (Table 1). Moreover, CA3 and CN5 promoted the growth and paramylon production of *E. gracilis* in C-NH_4_ + glucose without l-tryptophan (Figure 2 and Figure 3). Thus, the growth-promoting factors of CA3 and CN5 must differ from that of IAA. By contrast, the EPS-producing *Pseudoalteromonas* sp. MEBiC 03485 promotes the growth and paramylon production of *E. gracilis*, and supplementation with its free EPS at 133 mg L^−1^ or 333 mg L^−1^ significantly promoted the biomass and paramylon production of *E. gracilis* [29]. In the present study, CA3 produced free EPS in C-NH_4_ + glucose medium at concentrations of 26.7 ± 0.03 mg sugar L^−1^ and 16.5 ± 0.85 mg protein L^−1^ at pH 4.5 and 35.1 ± 0.08 mg sugar L^−1^ and 17.1 ± 0.11 mg protein L^−1^ at pH 7.5. Similarly, CN5 also produced free EPS in C-NH_4_ + glucose medium at concentrations of 31.7 ± 0.02 mg sugar L^−1^ and 17.1 ± 0.06 mg protein L^−1^ at pH 7.5 (Table 1). Furthermore, CA3 and CN5 produced EPS and attached to the surface of *E. gracilis* cells via the EPS matrix (Figure 4). Thus, EPS produced by CA3 and CN5 can potentially promote the growth and paramylon production of *E. gracilis*. However, the detected concentration of EPS produced by CA3 and CN5 was much lower than the previously reported effective EPS concentrations (133 and 333 mg L^−1^), suggesting the possibility that CA3 and CN5 might produce major factors other than EPS that are responsible for promoting growth and paramylon production.

These results clearly indicate that the growth- and paramylon-production-promoting factors of CA3 and CN5 are related to glucose metabolism or cell growth (Figure 2 and Figure 3). For example, the promoting factors might be produced through glucose metabolism or glucose-dependent growth to a high cell density and thereby promote *E. gracilis* cell growth and paramylon production. By contrast, CA3 and CN5 did not promote chlorophyll synthesis (Figure 2 and Figure 3). Chlorophylls are essential and limiting factors for photosynthesis [42] and also important in mixotrophic cultivation of *E. gracilis* [3]. Thus, the promoting factors of CA3 and CN5 might act more strongly in a heterotrophic mode than in an autotrophic mode under mixotrophic cultivation, which consists of autotrophic and heterotrophic growth modes. However, we did not test the effect of the bacteria under *E. gracilis* heterotrophic culture conditions.

In this study, we did not identify the factors or mechanisms by which CA3 and CN5 promote *E. gracilis* growth and paramylon production under mixotrophic conditions with glucose. MGPB can promote microalgal growth in various ways. Additionally, Zhu and Wakisaka [43] showed that the addition of ferulic acid made from rice promotes the growth and paramylon production of *E. gracilis*. Exogenous phytohormones, cytokinins, and abscisic acid can also promote the growth of *E. gracilis* [44]. Thus, it is apparent that various factors have the potential to promote the growth and paramylon production of *E. gracilis,* and further studies are needed to clarify the mechanisms whereby CA3 and CN5 promote these activities in *E. gracilis*.

After co-culturing *E. gracilis* cells with CA3 for 7 d under mixotrophic conditions with glucose, the biomass and paramylon productivity of *E. gracilis* increased by 1.8- and 3.5-fold, respectively, at pH 4.5 and by 1.9- and 3.5-fold, respectively, at pH 7.5 and relative to sterile *E. gracilis* cultures (Table 2). Additionally, the biomass and paramylon productivity of *E. gracilis* increased by 2.0- and 4.1-fold, respectively, at pH 7.5 when co-cultured with CN5 as compared with sterile *E. gracilis* culture (Table 2). Although the cultivation conditions differed, the promoting effects of CA3 and CN5 were higher than those (a 17–23% increase in biomass and a 25–35% increase in paramylon production) reported previously for *V. natriegens* [28], EPS-producing *Pseudoalteromonas* sp. MEBiC 03485 [29], and *Pseudoalteromonas* sp. MEBiC 03607 [30]. Therefore, CA3 and CN5 represent promising and useful MGPBs for increasing both biomass and paramylon yields of *E. gracilis* under mixotrophic conditions. Furthermore, the ability to cultivate *E. gracilis* under acidic conditions with CA3 provides an important advantage in terms of preventing microalgal and bacterial contamination, enabling large-scale cultivation of *E. gracilis*. Moreover, the attachment of CA3 and CN5 to the surface of *E. gracilis* flagella suggests that they might form a durable symbiotic association with *E. gracilis*. However, for the biotechnological application of CA3 and CN5, the method of co-culturing *E. gracilis* with these strains needs to be optimized in further studies. These findings demonstrated that our screening and isolation methods enabled the construction of an effective and practical *E. gracilis*–MGPB-association system for increasing the paramylon yield of *E. gracilis*.

## Figures and Tables

**Figure 1 microorganisms-09-01496-f001:**
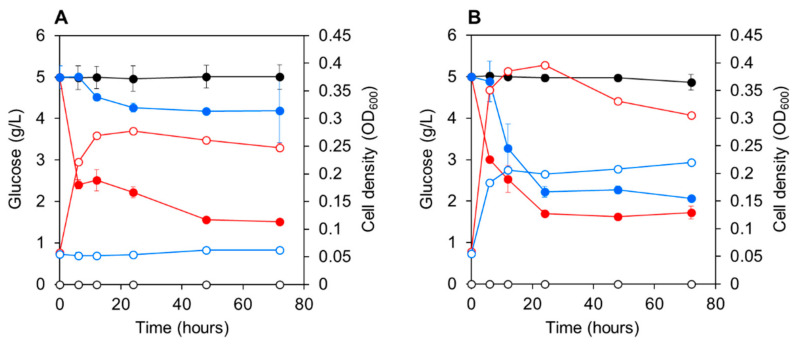
Glucose consumption and growth of *Enterobacter* sp. CA3 and *Emticicia* sp. CN5 cells at pH 4.5 (**A**) and 7.5 (**B**). Closed circles: glucose concentrations in CA3 cultures (red), CN5 cultures (blue), and bacteria-free control cultures (black). Open circles: cell densities (OD_600_) of CA3 cultures (red), CN5 cultures (blue), and bacteria-free control cultures (black). Data shown represent means ± SD (*n* = 3).

**Figure 2 microorganisms-09-01496-f002:**
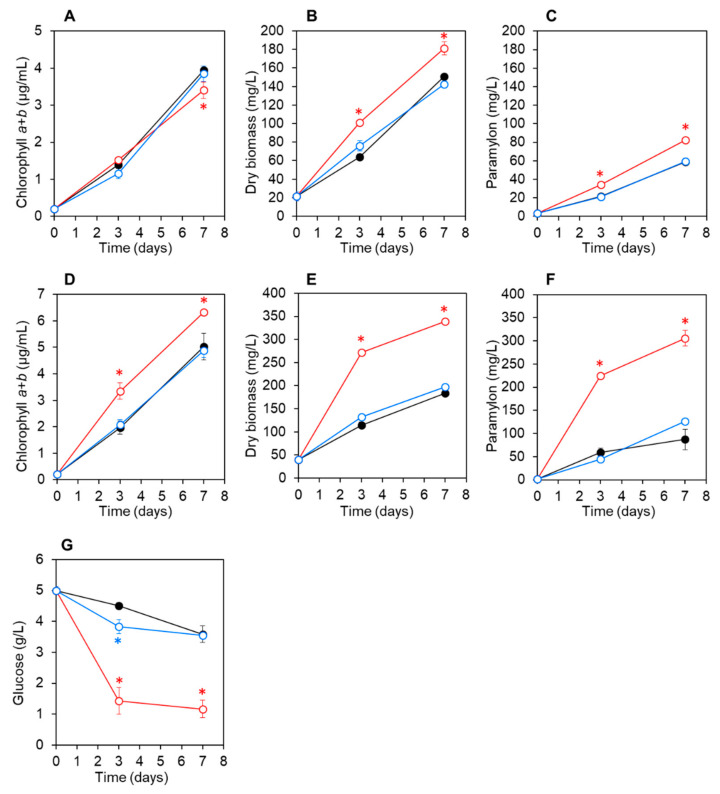
Changes in chlorophyll, paramylon, and glucose concentrations and biomass of *Euglena gracilis* cultured at pH 4.5 under autotrophic conditions without glucose (**A**–**C**) and under mixotrophic conditions with 5 g L^−1^ glucose (**D**–**G**). Open red circles: *E. gracilis* co-cultured with *Enterobacter* sp. CA3; open blue circles: *E. gracilis* co-cultured with *Emticicia* sp. CN5; and closed black circles: *E. gracilis* axenic control cultures. Data shown represent means ± SD (*n* = 3). Asterisks indicate significant difference (*p* < 0.05) between the *E. gracilis* co-culture with CA3 or CN5 and the *E. gracilis* axenic control culture.

**Figure 3 microorganisms-09-01496-f003:**
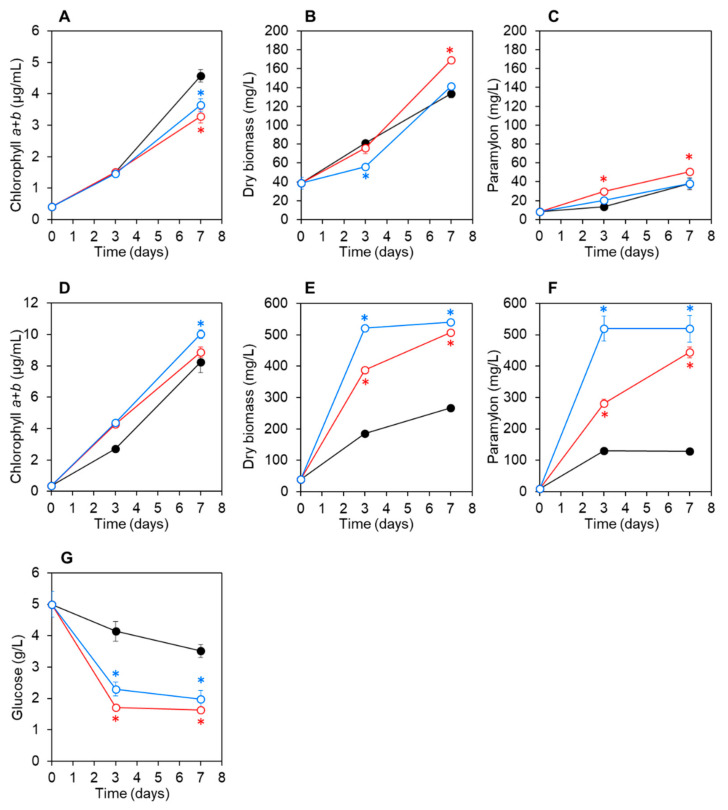
Changes in chlorophyll, paramylon, and glucose concentrations and biomass of *Euglena gracilis* cultured at pH 7.5 under autotrophic conditions without glucose (**A**–**C**) and under mixotrophic conditions with 5 g L^−1^ glucose (**D**–**G**). Open red circles: *E. gracilis* co-cultured with *Enterobacter* sp. CA3; open blue circles: *E. gracilis* co-cultured with *Emticicia* sp. CN5; and the closed black circles: *E. gracilis* axenic control cultures. Data shown represent means ± SD (*n* = 3). Asterisks indicate significant difference (*p* < 0.05) between the *E. gracilis* co-culture with CA3 or CN5 and the *E. gracilis* axenic control culture.

**Figure 4 microorganisms-09-01496-f004:**
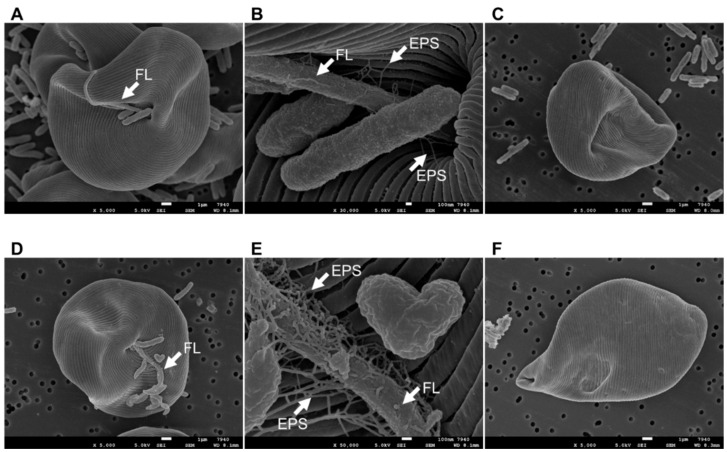
Scanning electron microscopy images of *Euglena gracilis* cells after 7 d of co-culture with *Enterobacter* sp. CA3 without washing (**A**,**B**) or after washing (**C**) or after 7 d of co-culture with *Emticicia* sp. CN5 without washing (**D**,**E**) or after washing (**F**). The FL arrows in panels (**A**,**B**,**D**,**E**) point to the flagella of *E. gracilis*. The EPS arrows in panels (**B**,**E**) point to extracellular polymeric substance matrix. The white scale bars in panels (**A**,**C**,**D**,**F**) represent 1 μm, and the scale bars in panels (**B**,**E**) represent 100 nm.

**Table 1 microorganisms-09-01496-t001:** Indole-3-acetic acid (IAA)-producing and extracellular polymeric substance (EPS)-producing activities of *Enterobacter* sp. CA3 and *Emticicia* sp. CN5 strains isolated in this study.

Strain and pH Condition	IAA-Producing Activity		EPS-Producing Activity			
	IAA Concentration with l-Tryptophan (μmol L^−1^) ^1^	IAA Concentration without l-Tryptophan	Free EPS		Cell-Bound EPS	
			Sugar (mg L^−1^) ^1^	Protein (mg L^−1^) ^1^	Sugar (mg L^−1^) ^1^	Protein (mg L^−1^) ^1^
CA3 at pH 4.5	5.48 ± 0.24	Negative	26.7 ± 0.03	16.5 ± 0.85	1.22 ± 0.00	1.07 ± 0.21
CA3 at pH 7.5	8.11 ± 0.23	Negative	35.1 ± 0.08	17.1 ± 0.11	2.01 ± 0.00	2.49 ± 0.24
CN5 at pH 4.5	NT ^2^	NT ^2^	NT ^2^	NT ^2^	NT ^2^	NT ^2^
CN5 at pH 7.5	2.51 ± 0.07	Negative	31.7 ± 0.02	17.1 ± 0.06	2.14 ± 0.00	2.54 ± 0.32

^1^ All values are presented as the mean ± SD (*n* = 3); ^2^ NT indicates a condition that was not tested because CN5 could not be grown at pH 4.5.

**Table 2 microorganisms-09-01496-t002:** Biomass and paramylon productivities of *Euglena gracilis* cells co-cultured with *Enterobacter* sp. CA3 or *Emticicia* sp. CN5 and axenic control cultures under various conditions.

Culture Condition		Co-Culture Condition	Final Biomass and Paramylon Concentration after 7 d	
			Biomass (mg L^−1^) ^1^	Paramylon (mg L^−1^) ^1^
pH 4.5	Autotrophic	Co-culture with CA3	181.3 ± 7.1 (1.2-fold) ^2^	82.5 ± 3.5 (1.4-fold)
		Co-culture with CN5	142.7 ± 1.3 (0.95-fold)	59.4 ± 2.8 (1.0-fold)
		Control	150.7 ± 0.7	58.9 ± 2.3
	Mixotrophic with glucose	Co-culture with CA3	338.7 ± 4.2 (1.8-fold)	305.7 ± 16.8 (3.5-fold)
		Co-culture with CN5	197.3 ± 4.8 (1.1-fold)	126.3 ± 1.3 (1.4-fold)
		Control	184.0 ± 5.3	187.1 ± 22.4
pH 7.5	Autotrophic	Co-culture with CA3	169.3 ± 0.2 (1.3-fold)	50.6 ± 1.1 (1.3-fold)
		Co-culture with CN5	141.3 ± 2.3 (1.1-fold)	37.9 ± 5.0 (1.0-fold)
		Control	133.3 ± 4.2	37.9 ± 6.5
	Mixotrophic with glucose	Co-culture with CA3	508.0 ± 1.1 (1.9-fold)	443.6 ± 17.7 (3.5-fold)
		Co-culture with CN5	540.0 ± 2.8 (2.0-fold)	518.9 ± 41.7 (4.1-fold)
		Control	266.7 ± 3.1	128.0 ± 2.0

^1^ All values are presented as the mean ± SD (*n* = 3); ^2^ The number in parentheses indicates the ratio of each value as compared with each control experimental value (co-culture with CA3/control or co-culture with CN5/control).

## Data Availability

The 16S rRNA sequence data (1431 bp) of CA3 and CN5 were submitted to the DDBJ/EMBL/GenBank databases under accession numbers LC604062 and LC604063, respectively.

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
