# Peer review of "Isolation and Characterization of *Euglena gracilis*-Associated Bacteria, *Enterobacter* sp. CA3 and *Emticicia* sp. CN5, Capable of Promoting the Growth and Paramylon Production of *E. gracilis* under Mixotrophic Cultivation"

_microorganisms, 2021, doi:10.3390/microorganisms9071496_

Round 1

Reviewer 1 Report

This manuscript focuses on the use of bacteria as a promising strategy to improve Euglena gracilis biomass and paramylon production. Among the 23 species/strains isolated from a sewage effluent, 2 of them – Enterobacter sp. CA3 and Emticicia sp. CN5 – proved to considerably enhance the growth of microalgae and production of paramylon under mixotrophic conditions. Although the topic is interesting and the manuscript is well-structured, some minor revisions are needed before accepting the paper. Therefore, I have some comments to help the authors to improve the manuscript.

1) A thorough revision of the manuscript should be done so that English level could be improved;

2) Could you please clarify the reason why you use 3 culture media (C, CYP, and C-NH4) in E. gracilis pre-cultures and then only C-NH4 is used in co-cultures and CSL assays?

3) How did you ensure that E. gracilis inocula were similar if you inoculated co-cultures only based on the volume and not on OD, as done for the bacteria?

4) Pictures from Figure S2 are a bit blurred. If possible, please provide pictures with higher quality.

5) Do you have any idea why both Enterobacter sp. CA3 and Emticicia sp. CN5 strains were not able to produce IAA in the absence of L-tryptophan? It could be important to provide that information.

6) Page 11, line 438 – Which biomass are you referring to? The biomass from E. gracilis, correct? If so, please rephrase and make it clear.

Author Response

Thank you very much for reviewing our manuscript and offering valuable comments. We deeply appreciate your careful consideration for our manuscript. According to your comments, we have carefully checked the language of the manuscript to ensure a high level of readability. Our manuscript received English proofreading by Editage Japan. Also, we revised our manuscript. The modified parts were colored in blue.

We are very happy if you kindly consider that our revision is enough to meet your comments.

Response 1:

Our manuscript received English proofreading by Editage Japan.

Response 2:

In this study, we used CYP for subculture of E. gracilis. CYP is recommended for culture of E. gracilis by the Microbial Culture Collection at the National Institute for Environmental Studies. CYP includes organic compounds. Also, E. gracilis use ammonium as nitrogen. Therefore, we used C-NH4 in co-culture experiments. We explained the above answers in revision.

Response 3:

We confirmed the E. gracilis inoculum concentration as biomass concentration (mg/L). We added the explanation in revision.

Response 4:

As you pointed out, quality of Figure S2 are a bit blurred. However, unfortunately, this is the limit of camera performance.

Response 5:

Thank you for your suggestion. L-tryptophan is a main precursor for IAA biosynthesis pathways in bacteria. Therefore, Enterobacter sp. CA3 and Emticicia sp. CN5 strains produced IAA in the presence of L-tryptophan. We explained the above in revised manuscript.

Response 6

As pointed out, the biomass was unclear. We revised as E. gracilis biomass in the revision.

Reviewer 2 Report

Interesting and well executed manuscript

Author Response

Thank you very much for reviewing our manuscript and comments.

We have carefully checked the language of the manuscript to ensure a high level of readability. Our manuscript received English proofreading by Editage Japan.

Reviewer 3 Report

Brief summary
The authors aimed to isolate, identify and characterize sewage-effluent bacteria that are associated with Euglena gracilis in a presumably symbiotic relationship, which promote growth and paramylon production under mixotrophic cultivation conditions at acidic or neutral pH .

Broad comments
The strengths of this manuscript are its aims, the introduction, and the characterization of the isolated bacterial strains. The co-cultivation experiments are the core of this study and form the basis for the most important conclusions. However, most of the growth curves are incomplete. This means that the presented data cannot support the drawn conclusions sufficiently. Further weaknesses are a one-sided comparison of the main results with the literature in the discussion and minor flaws in the methodology.

Specific comments 

1. Introduction
This section is well-written.

1. L 50-54: Cultivation under acidic conditions mostly prevents bacterial contamination, microalgae are less of an issue. However, it does not prevent fungal contaminations, so it cannot enable open-pond, large-scale cultivation without further process optimization (e.g., use of antifungal agents), which could proof cost-inhibiting. Contaminations are also an issue in closed system though, so either the issue of fungal contamination needs to be mentioned or "open-pond" removed. 

2. Materials and Methods
This section should be shortened by referring to published protocols (applies to most of the subsections), which would also clarify whether a described method was developed for this study or has already been published.

2. L 133: Why was corn steep liquor used as C-source and not glucose? This seems to be an unnecessary inconsistency in the methodology.

3. L 136-143: The co-culture should have been passaged several times to enrich Euglena-associated bacteria. It seems rather lucky that the desired strains were isolated at all. 

4. L 154: Usually lower speeds are used (~5,000 × g) for centrifuging bacteria to maximize the viability of the pelleted cells.

5. L 158: This incubation time seems arbitrary and is a major issue because it is the reason for the mentioned missing data points.

6. L 182: Usually higher speeds are used (≥20,000 × g) for obtaining bacteria-free supernatants.

7. L 234: PBS or another buffer should have been used for washing as osmotic stress can damage microscopy samples and create artifacts.

8. L 254: The incubation time seems rather short and the temperature unnecessarily high. Usually, the dry weight of microalgae is obtained by drying the cell mass at a lower temperature (e.g., 70°C) overnight or "until dry", or by freeze-drying.

3. Results
This section needs to be amended with further experimental data. 

9. L 290-298: Was this experiment performed in biological replicate? If not, the stated values are only qualitative and should be moved to the Supplementary Materials, and the sentence L 294-296 needs to be removed.

10. L 294-295: The repetition of the exact cultivation conditions is confusing and needs to be removed either way. 

11. L 371-395/Figure 2: Most of the cultures had not reached stationary phase, which means that only snapshots of the respective growth curves are presented here. Also, the cultures "E. gracilis co-cultured with Emticicia sp. CN5" as well as "E. gracilis axenic control" still contained substantial amounts of unused glucose. It is possible that the observed differences in cell mass and paramylon productivity were due to differences in the respective growth rates of the cultures and, therefore, only temporary. Productivities of cultures in batch cultivations can only be compared during the same growth phase or after the entire growth curve has been recorded to determine the maximum productivity/yield. Therefore, these growth curves need to be amended with more data points. 

12. L 385-404/Table 2: These observations and values cannot be compared (see comment 11), so they cannot be used to draw meaningful conclusions. Also, the biomass and paramylon productivities (even those of cultures that had arguably reached stationary phase such as "E. gracilis co-cultured with 382 Enterobacter sp. CA3") were very low compared to values from the literature, which indicates that the cultivation conditions were suboptimal. If that is the case, the entire study loses credibility, since it is not clear if the observed growth and paramylon production-enhancing effects will also occur when optimized cultivation protocols are used.

13. L 400-440/Table 2/Figure 3: The same criticism as in comments 11 and 12 applies here. Even tough the co-cultures apparently had reached the stationary phase, the axenic control had not, which means that these growth curves also need to be amended.

Discussion
The discussion is mostly exhaustive, but it is also repeating too much information already stated in the introduction and should be shortened (short summary of the most important findings, comparisons with the literature, and a short outlook only). On the other hand, the main results (co-cultivation experiments) are discussed rather one-sided. Also, it remains to be seen how exactly the discussion needs to be modified after further experimental data (see comments 11-13) have been incorporated in this manuscript.

14. L 462-465: This sentence is already in the introduction and should be replaced or removed.

15. L 492-494: Further possible explanation: The bacterial strains were probably starving under autotrophic cultivation conditions because Euglena does not supply a C-source for its bacterial symbionts.

16. L 529-531: This sentence is already in the introduction and should be replaced or removed.

17. L 544-547: The results of the core experiments need to be discussed in the context of all relevant reports. As mentioned earlier (see comment 12), the growth and paramylon productivities (i.e., yields) reported here are very low, especially under mixotrophic conditions, and it could turn out that the observed prouctivity-promoting effects are insignificant or negligible after the cultivation conditions have been optimized according to published protocols. However, the identification and characterization of Euglena-associated bacteria is still very interesting from a basic research point of view, even if there is no immediate biotechnological application, or as a starting point for further optimization.

18. L 548 -540: Again, repetitive, needs to be removed or rephrased. Also, see comment 1.

Author Response

Thank you very much for reviewing our manuscript and offering valuable comments. We deeply appreciate your careful consideration for our manuscript. According to your comments, we have carefully checked and revised our manuscript. The modified parts were colored in blue.

We are very happy if you kindly consider that our revision is enough to meet your comments.

Response 1:

Thank you for your suggestion. We added the information on bacterial contamination and removed “open-pond”.

Response 2-1:

Thank you for your comments. We added the citation to published protocols at IAA production test, EPS production test, phenol-sulfuric acid method chlorophyll measurement. However, we conducted these tests and methods with some modification. So we added the additional explanations in each protocol. Other methods were not published protocols and were developed in this study.

Response 2-2:

As pointed out, there were no information on why corn steep liquor was used as C-souse. We added the reason in revised text.

Response 3:

As pointed out, in this study, Euglena-sewage bacteria co-culture have been passaged several times to build a stronger relationship Euglena and bacteria and to enrich Euglena-associated bacteria. During these enrichment co-culture, Euglena growth was promoted.

Response 4:

In our laboratory, we collected bacterial cells by centrifugation at 10,000 g. We had checked the bacterial growth and activity collected by the centrifugation.

Response 5:

In this study, we cultured Euglena for 7 days. As you pointed out, Euglena growth trended to increase at 7 days. In this study, we aimed to evaluate “promoting the growth and paramylon production”. Promoting means accelerating. Therefore, we evaluate the growth and paramylon production within 7 days. We modified the revised manuscript as following; “In a 7-day E. gracilis mixotrophic culture with glucose, CA3 increased E. gracilis biomass and paramylon production 1.8-fold and 3.5-fold, respectively (at pH 4.5), or 1.9-fold and 3.5-fold, respectively (at pH 7.5).”, “The growth- and paramylon-production-promoting effects of CA3 and CN5 were evaluated for comparison with E. gracilis axenic control culture in 7-d cultures”, and “Both bacterial strains were capable of significantly promoting the growth and paramylon production of E. gracilis at their respective pH values during a 7-d cultivation”.

In further studies, we would evaluate the culture period of co-culture.

Response 6:

In our laboratory, we collected bacterial cells and collected bacteria-free supernatant by centrifugation at 10,000 g. We had checked that the supernatant did not contain bacteria.

Response 7:

As pointed out, PBS or other buffers should have been used for washing. In previous study, we had used distilled water for washing and had checked that there are no negative effect of distilled water washing.

Response 8:

For dry-weight determination, we dried Euglena cell at 90°C for 3 h. We had confirmed that Euglena cells were already dried by the above temperature and time (the weight were not changed at 90°C for 3 h and 24h).

Thank you for your comment. If we monitor temperature-sensitive materials, we will dried Euglena at lower temperature or freeze-drying.

Response 9:

We are so sorry. Our explanation was insufficient. We conducted the cultivation in triplicate. We added the explanation at section 2.3.

Response 10:

Thank you for your comment. We removed the overlapping information. And, we added the information and data on Supplemental file Figure S1.

Response 11:

As pointed out, Euglena growth had not reached stationary phase. In this study, we evaluated the growth and paramylon production within 7 days. We compared the growth and paramylon production of “Euglena co-cultured with CA3 or CN5” to “Euglena axenic control” within 7 days at same condition. We modified the revised manuscript as following; “In a 7-day E. gracilis mixotrophic culture with glucose, CA3 increased E. gracilis biomass and paramylon production 1.8-fold and 3.5-fold, respectively (at pH 4.5), or 1.9-fold and 3.5-fold, respectively (at pH 7.5).”, “The growth- and paramylon-production-promoting effects of CA3 and CN5 were evaluated for comparison with E. gracilis axenic control culture in 7-d cultures”, and “Both bacterial strains were capable of significantly promoting the growth and paramylon production of E. gracilis at their respective pH values during a 7-d cultivation”.

Thank you for your understanding.

Response 12:

Like response 11, in this study, we evaluated the growth and paramylon production within 7 days. We compared the growth and paramylon production of “Euglena co-cultured with CA3 or CN5” to “Euglena axenic control” within 7 days at same condition.

In further studies, we would like to conduct experiments under optimized conditions and for longer term.

Response 13:

Like responses 11 and 12, in this study, we evaluated the growth and paramylon production within 7 days. We compared the growth and paramylon production of “Euglena co-cultured with CA3 or CN5” to “Euglena axenic control” within 7 days at same condition.

In further studies, we would like to conduct experiments under optimized conditions and for longer term.

Response 14-1:

Thank you for your suggestion. We have shortened discussion part and delete the overlapping parts. However, we left some descriptive texts to support the reader's understanding.

Response 14-2:

We removed the sentences.

Response 15:

Thank you for providing the additional possible explanations. We also consider the effects of Euglena on bacteria are important. Euglena might support the bacterial growth and activity. This study is a basic work on the Euglena growth and paramylon production promotion by associated bacteria CA3 and CN5. Isolation and basic-characterization of CA3 and CN5 were the core-findings and significant points of this work.

In the next study, we will investigate the relationship between Euglena and CA3/CN5 more in detailed. Thank you for your suggestion.

Response 16:

We removed the repeating information.

Response 17:

Thank you for your comments. As pointed out, the identification and characterization of Euglena-associated bacteria are the core findings and significant points of this study. For biotechnological application of CA3 and CN5, we have to optimize the co-culture method of Euglena and the strains in further studies. We added the comment in revised manuscript.

Response 18:

We removed “open-pond”.

Reviewer 4 Report

The manuscript submitted by Rubiyatno and coworkers deals with the evaluation of a heteroxenic culture of Euglena gracilis associated to other microorganisms and the improved yield in paramylon production.

The presentation is qualified and I am sure will be of the interest of Microorganism readers.  I have some concerns about the manuscript I would like to ask the authors before giving my final opinion.

Line 72.- I do not understand the relationship between bacterial production of IAA and EPS and the increasing synthesis of paramylon. I consider it would be interesting to complement the introduction of the manuscript with some information about the influence of these compounds on carbohydrate metabolism, UDPGlucose synthesis or paramylon synthase activity.

Line 133.- Why CSL is used to growth Euglena gracilis and not glucose? It would be nice to include an explanation of the benefits of its use.

Line 294.- The biomass of E. gracilis grown in C-NH4 with 0.5 g L 294 ‒1 CSL, with or without effluent bacteria at pH 4.5 and pH7.5 are showed into results. The biomass is higher in heteroxenic culture than in the axenic culture of Euglena. Which is the aim of this presentation? Is it no obvious?

Line 401.- Table 2 shows the results of paramylon production in Euglena cultures together with bacteria. They obtain a paramylon yield of 90% of total biomass. Did you carry out controls in the determination with phenol - sulfuric acid but on bacterial biomass alone?

Author Response

Thank you very much for reviewing our manuscript and offering valuable comments. We deeply appreciate your careful consideration for our manuscript. We have revised our manuscript. The major modified parts were colored in blue.

We are very happy if you kindly consider that our revision is enough to meet your comments.

Response 1:

Thank you for your suggestion. The relationship between bacterial production of IAA and EPS and the increasing synthesis of paramylon are still unclear. The previous studies revealed that the expression of gene related to paramylon synthase was increased by EPS producing bacteria. Therefore, we explained the above comment in revised manuscript.

Response 2:

As you pointed out, why CSL was used to growth E. gracilis was unclear. We explained the reason in revised manuscript.

Response 3:

We wanted to state that the sewage effluent bacteria promoted the growth of E. gracilis at both pH conditions. Therefore, we modified in revised manuscript.

Response 4:

Yes, we also think that paramylon yield of 90% of total biomass was so high. In our study, we carefully measured paramylon content in triplicate. And we determined the paramylon concentration in bacterial biomass alone control test. We did not detect paramylon in bacteria alone control test. Therefore, CA3 and CN5 could not produce paramylon. We explained the above in revised manuscript.

Round 2

Reviewer 3 Report

Dear authors,

Thank you for addressing my comments in detail. The manuscript is acceptable for publication in its present form. I am looking forward to hear more from this line of enquiry in the future.

Good luck and best regards

Reviewer 4 Report

I recommend accepting the manuscript in its present form for being published in Microorganism journal.

Regards